# Projection Regret: Reducing Background Bias for Novelty Detection via Diffusion Models

**Sungik Choi**[1]    **Hankook Lee**[1]    **Honglak Lee**[1]    **Moontae Lee**[1,2]
[1]LG AI Research    [2]University of Illinois Chicago
{sungik.choi, hankook.lee, honglak.lee, moontae.lee}@lgresearch.ai

## Abstract

Novelty detection is a fundamental task of machine learning which aims to detect abnormal (*i.e.* out-of-distribution (OOD)) samples. Since diffusion models have recently emerged as the de facto standard generative framework with surprising generation results, novelty detection via diffusion models has also gained much attention. Recent methods have mainly utilized the reconstruction property of in-distribution samples. However, they often suffer from detecting OOD samples that share similar background information to the in-distribution data. Based on our observation that diffusion models can *project* any sample to an in-distribution sample with similar background information, we propose *Projection Regret (PR)*, an efficient novelty detection method that mitigates the bias of non-semantic information. To be specific, PR computes the perceptual distance between the test image and its diffusion-based projection to detect abnormality. Since the perceptual distance often fails to capture semantic changes when the background information is dominant, we cancel out the background bias by comparing it against recursive projections. Extensive experiments demonstrate that PR outperforms the prior art of generative-model-based novelty detection methods by a significant margin.

## 1 Introduction

Novelty detection [1], also known as out-of-distribution (OOD) detection, is a fundamental machine learning problem, which aims to detect abnormal samples drawn from far from the training distribution (*i.e.*, in-distribution). This plays a vital role in many deep learning applications because the behavior of deep neural networks on OOD samples is often unpredictable and can lead to erroneous decisions [2]. Hence, the detection ability is crucial for ensuring reliability and safety in practical applications, including medical diagnosis [3], autonomous driving [4], and forecasting [5].

The principled way to identify whether a test sample is drawn from the training distribution $p_{\text{data}}(\mathbf{x})$ or not is to utilize an explicit or implicit *generative model*. For example, one may utilize the likelihood function directly [6] or its variants [7, 8] as an OOD detector. Another direction is to utilize the generation ability, for example, reconstruction loss [9] or gradient representations of the reconstruction loss [10]. It is worth noting that the research direction of utilizing generative models for out-of-distribution detection has become increasingly important in recent years as generative models have been successful across various domains, e.g., vision [11, 12] and language [13], but also they have raised various social issues including deepfake [14] and hallucination [15].

Among various generative frameworks, diffusion models have recently emerged as the most popular framework due to their strong generation performance, e.g., high-resolution images [16], and its wide applicability, e.g., text-to-image synthesis [11]. Hence, they have also gained much attention as an attractive tool for novelty detection. For example, Mahmood et al. [17] and Liu et al. [18] utilize the reconstruction error as the OOD detection metric using diffusion models with Gaussian noises [17] or checkerboard masking [18]. Somewhat unexpectedly, despite the high-quality generation results,

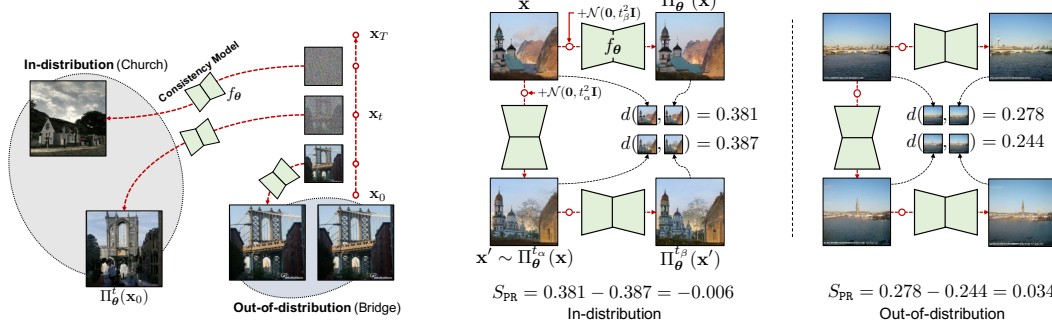

(a) Projection $\Pi_{\boldsymbol{\theta}}$          (b) Projection Regret $S_{\mathrm{PR}}$

Figure 1: **Conceptual illustration of our framework. (a)** Given an image $\mathbf{x}_0$, the projection of the image, $\Pi_{\boldsymbol{\theta}}^t(\mathbf{x}_0)$, is defined by diffusing $\mathbf{x}_0$ until an intermediate timestep $t$ and then reversing the process using the consistency model $f_{\boldsymbol{\theta}}$. The projection $\Pi_{\boldsymbol{\theta}}^t$ can map an out-of-distribution image to an in-distribution one with similar backgrounds when $t$ is neither too early nor too late. **(b)** Our proposed method, Projection Regret, computes the abnormality score $S_{\mathrm{PR}}$ by capturing the semantic difference between a test image and its projection. We further utilize the two-step projection to cancel out the background bias, especially for the case when background information is dominant.

their detection performance still lags behind other generative models, *e.g.*, energy-based models [19]. We find that these undesired results are related to the background bias problem: the reconstruction error is often biased by non-semantic (*i.e.*, background) information. Namely, the prior methods often struggle to detect out-of-distribution images with similar backgrounds (*e.g.*, CIFAR-100 images against CIFAR-10 ones).

**Contribution.** To reduce the background bias, we first systemically examine the behavior of diffusion models. We observe that the models can change semantic information to in-distribution without a major change to background statistics by starting from an intermediate point of the diffusion process and reversing the process, which is referred to as *projection* in this paper (see Figure 1a). Based on this observation, we suggest using the perceptual distance [20] between a test image and its projection to detect whether the test image is out-of-distribution.

Although our projection slightly changes the background statistics, the perceptual distance sometimes fails to capture semantic differences when the background information is dominant over the semantics. To resolve this issue, we propose *Projection Regret*, which further reduces the background bias by canceling out the background information using recursive projections (see Figure 1b). We also utilize an ensemble of multiple projections for improving detection performance. This can be calculated efficiently via the consistency model [21], which supports the reverse diffusion process via only one network evaluation.

Through extensive experiments, we demonstrate the effectiveness of Projection Regret (PR) under various out-of-distribution benchmarks. More specifically, PR outperforms the prior art of diffusion-based OOD detection methods with a large margin, *e.g.*, $0.620 \rightarrow 0.775$ detection accuracy (AUROC) against a recent method, LMD [18], under the CIFAR-10 *vs* CIFAR-100 detection task (see Table 1). Furthermore, PR shows superiority over other types of generative models (see Table 2), including EBMs [22, 23, 24] and VAEs [25]. We also propose a perceptual distance metric using underlying features (*i.e.*, feature maps in the U-Net architecture [26]) of the diffusion model. We demonstrate that Projection Regret with this proposed metric shows comparable results to that of LPIPS [20] and outperforms other distance metrics, *e.g.*, SSIM [27] (see Table 5).

To sum up, our contributions are summarized as follows:

- We find that the perceptual distance between the original image and its projection is a competitive abnormality score function for OOD detection (Section 3.1).

- We also propose Projection Regret (PR) that reduces the bias of non-semantic information to the score (Section 3.2) and show its efficacy across challenging OOD detection tasks (Section 4.2).

- We also propose an alternative perceptual distance metric that is computed by the decoder features of the pre-trained diffusion model (Section 4.3).

## 2 Preliminaries

### 2.1 Problem setup

Given a training distribution $p_{\text{data}}(\mathbf{x})$ on the data space $\mathcal{X}$, the goal of out-of-distribution (OOD) detection is to design an abnormality score function $S(\mathbf{x}) \in \mathbb{R}$ which identifies whether $\mathbf{x}$ is drawn from the training distribution (*i.e.*, $S(\mathbf{x}) \leq \tau$) or not (*i.e.*, $S(\mathbf{x}) > \tau$) where $\tau$ is a hyperparameter. For notation simplicity, "A *vs* B" denotes an OOD detection task where A and B are in-distribution (ID) and out-of-distribution (OOD) datasets, respectively. For evaluation, we use the standard metric, area under the receiver operating characteristic curve (AUROC).

In this paper, we consider a practical scenario in a generative model $p_\theta(\mathbf{x})$ pre-trained on $p_{\text{data}}(\mathbf{x})$ is only available for OOD detection. Namely, we do not consider (a) training a detector with extra information (e.g., label information and outlier examples) nor (b) accessing internal training data at the test time. As many generative models have been successfully developed across various domains and datasets in recent years, our setup is also widely applicable to real-world applications as well.

### 2.2 Diffusion and consistency models

**Diffusion models (DMs).** A *diffusion model* is defined over a diffusion process starting from the data distribution $\mathbf{x}_0 \sim p_{\text{data}}(\mathbf{x})$ with a stochastic differential equation (SDE) and its reverse-time process can be formulated as probability flow ordinary differential equation (ODE) [28, 29]:

$$\text{Forward SDE:} \quad d\mathbf{x}_t = \boldsymbol{\mu}(\mathbf{x}_t, t)dt + \sigma(t)d\mathbf{w}_t,$$

$$\text{Reverse ODE:} \quad d\mathbf{x}_t = \left[ \boldsymbol{\mu}(\mathbf{x}_t, t) - \tfrac{1}{2}\sigma(t)^2 \nabla \log p_t(\mathbf{x}_t) \right] dt,$$

where $t \in [0, T]$, $T > 0$ is a fixed constant, $\boldsymbol{\mu}(\cdot, \cdot)$ and $\sigma(\cdot)$ are the drift and diffusion coefficients, respectively, and $\mathbf{w}_t$ is the standard Wiener process. The variance-exploding (VE) diffusion model learns a time-dependent denoising function $D_{\boldsymbol{\theta}}(\mathbf{x}_t, t)$ by minimizing the following denoising score matching objective [29]:

$$\mathcal{L}_{\text{DM}} := \mathbb{E}_{t \sim \mathcal{U}[\epsilon, T], \mathbf{x} \sim p_{\text{data}}, \mathbf{z} \sim \mathcal{N}(\mathbf{0}, \mathbf{I})} \left[ \| D_{\boldsymbol{\theta}}(\mathbf{x} + \sigma_t \mathbf{z}, t) - \mathbf{x} \|_2^2 \right], \tag{1}$$

where $\sigma_t$ depends on $\sigma(t)$. Following the practice of Karras et al. [29], we set $\sigma_t = t$ and discretize the timestep into $\epsilon = t_0 < t_1 < ... < t_N = T$ for generation based on the ODE solver.

**Consistency models (CMs).** A few months ago, Song et al. [21] proposed *consistency models*, a new type of generative model, built on top of the consistency property of the probability flow (PF) ODE trajectories: points $(\mathbf{x}_t, t)$ on the same PF ODE trajectory map to the same initial point $\mathbf{x}_\epsilon$, *i.e.*, the model aims to learn a consistency function $f_{\boldsymbol{\theta}}(\mathbf{x}_t, t) = \mathbf{x}_\epsilon$. This function $f_{\boldsymbol{\theta}}$ can be trained by trajectory-wise distillation. The trajectory can be defined by the ODE solver of a diffusion model (*i.e.*, consistency distillation) or a line (*i.e.*, consistency training). For example, the training objective for the latter case can be formulated as follows:

$$\mathcal{L}_{\text{CM}} := \mathbb{E}_{i \sim \mathcal{U}\{0, ..., N-1\}, \mathbf{x} \sim p_{\text{data}}, \mathbf{z} \sim \mathcal{N}(\mathbf{0}, \mathbf{I})} \left[ d\big( f_{\boldsymbol{\theta}^-}(\mathbf{x} + \sigma_{t_i} \mathbf{z}, t_i), f_{\boldsymbol{\theta}}(\mathbf{x} + \sigma_{t_{i+1}} \mathbf{z}, t_{i+1}) \big) \right], \tag{2}$$

where $d$ is the distance metric, *e.g.*, $\ell_2$ or LPIPS [20], and $\boldsymbol{\theta}^-$ is the target parameter obtained by exponential moving average (EMA) of the online parameter $\boldsymbol{\theta}$. The main advantage of CMs over DMs is that CMs support one-shot generation. In this paper, we mainly use CMs due to their efficiency, but our method is working with both diffusion and consistency models (see Table 6).

## 3 Method

In this section, we introduce Projection Regret (PR), a novel and effective OOD detection framework based on a diffusion model. Specifically, we utilize diffusion-based *projection* that maps an OOD image to an in-distribution one with similar background information (Section 3.1) and design an abnormality score function $S_{\text{PR}}$ that reduces background bias using recursive projections (Section 3.2).

### 3.1 Projection

Although previous OOD detection methods based on generative models have been successfully applied to several benchmarks [8, 17], *e.g.*, CIFAR-10 *vs* SVHN, they still struggle to detect out-of-distribution samples of similar backgrounds to the in-distribution data (*e.g.*, CIFAR-10 *vs* CIFAR-100). We here examine this challenging scenario with diffusion models.

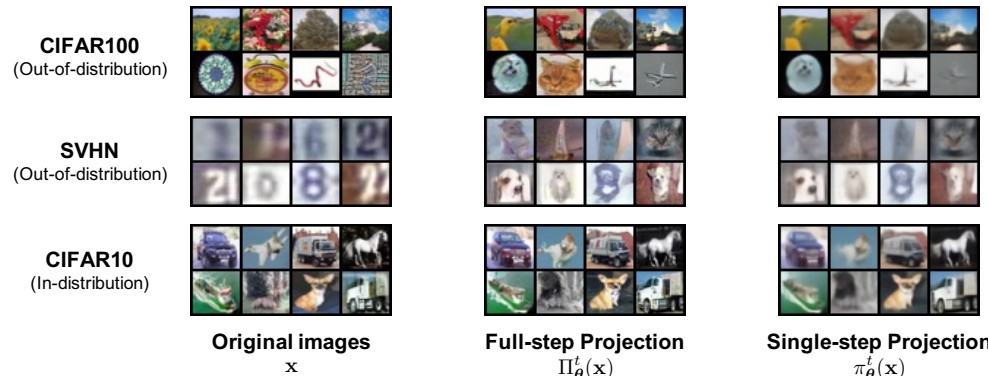

Figure 2: **Projection of the OOD and the in-distribution image.** We show full-step projection and single-step projection results for in-distribution (CIFAR-10) and OOD (CIFAR-100, SVHN) images.

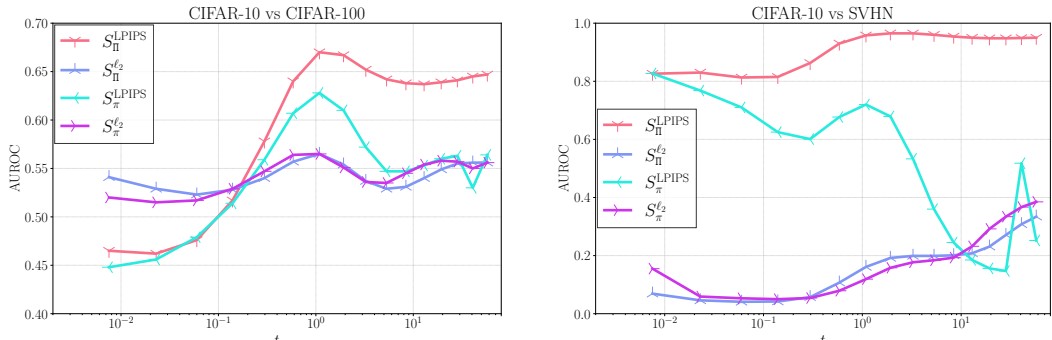

Figure 3: **AUROC of the OOD detection task under different design choices and time-steps.** **(Left):** CIFAR-10 vs CIFAR-100 OOD detection. **(Right):** CIFAR-10 vs SVHN OOD detection. Our proposed abnormality score **(red)** is the best choice.

Given discrete timesteps $\epsilon = t_0 < t_1 < \ldots < t_N = T$, we first suggest two projection schemes, $\pi_{\boldsymbol{\theta}}^{t_i}(\mathbf{x}, \mathbf{z})$ and $\Pi_{\boldsymbol{\theta}}^{t_i}(\mathbf{x}, \mathbf{z})$, by reversing the diffusion process until $t = t_{i-1}$ and $t = \epsilon$, respectively, starting from the noisy input $\mathbf{x}_{t_i} := \mathbf{x} + t_i\mathbf{z}$. Note that $\pi_{\boldsymbol{\theta}}$ can be implemented by only the diffusion-based denoising function $D_{\boldsymbol{\theta}}$ while $\Pi_{\boldsymbol{\theta}}$ can be done by multi-step denoising with $D_{\boldsymbol{\theta}}$ or single-step generation with the consistency model $f_{\boldsymbol{\theta}}$ as described in Section 2.2. We refer to $\pi_{\boldsymbol{\theta}}$ and $\Pi_{\boldsymbol{\theta}}$ as single-step and full-step projections, respectively, and the latter is illustrated in Figure 1a. We simply use $\pi_{\boldsymbol{\theta}}^{t_i}(\mathbf{x})$ and $\Pi_{\boldsymbol{\theta}}^{t_i}(\mathbf{x})$ with $\mathbf{z} \sim \mathcal{N}(\mathbf{0}, \mathbf{I})$ unless otherwise stated.

We here systemically analyze the projection schemes $\pi_{\boldsymbol{\theta}}^{t_i}$ and $\Pi_{\boldsymbol{\theta}}^{t_i}$ with the denoising function $D_{\boldsymbol{\theta}}$ and the consistency function $f_{\boldsymbol{\theta}}$, respectively, trained on CIFAR-10 [30] for various timesteps $t \in [\epsilon, T]$. Note that both projections require only one network evaluation. Figure 2 shows the projection results of CIFAR-100 [30] and SVHN [31] OOD images where $t_i \approx 1.09$ (*i.e.*, $i = 7$, $N = 17$, and $T = 80$). One can see semantic changes in the images while their background statistics have not changed much. Furthermore, the shift from the in-distribution CIFAR-10 dataset is less drastic than the OOD dataset. We also observe that the projections (i) do not change any information when $t_i$ is too small, but (ii) generate a completely new image when $t_i$ is too large, as shown in Figure 1a. Moreover, we observe that the single-step projection $\pi_{\boldsymbol{\theta}}^{t_i}$ outputs more blurry image than the full-step projection $\Pi_{\boldsymbol{\theta}}^{t_i}$.

Based on these observations, we propose projection-based OOD detection scores by measuring the reconstruction error between an input $\mathbf{x}$ and its projection with a distance metric $d(\cdot, \cdot)$:

$$S_{\pi}^d(\mathbf{x}, t; \boldsymbol{\theta}) := \mathbb{E}_{\mathbf{z} \sim \mathcal{N}(\mathbf{0}, \mathbf{I})} \left[ d\left(\mathbf{x}, \pi_{\boldsymbol{\theta}}^t(\mathbf{x}, \mathbf{z})\right) \right], \quad S_{\Pi}^d(\mathbf{x}, t; \boldsymbol{\theta}) := \mathbb{E}_{\mathbf{z} \sim \mathcal{N}(\mathbf{0}, \mathbf{I})} \left[ d\left(\mathbf{x}, \Pi_{\boldsymbol{\theta}}^t(\mathbf{x}, \mathbf{z})\right) \right]. \quad (3)$$

For the distance $d$, in this paper, we use the common metrics in computer vision: the $\ell_2$ distance and the perceptual distance, LPIPS [20]. One may expect that the latter is fit for our setup since it focuses on semantic changes more than background changes. Other choices are also studied in Section 4.3.

We report the AUROC on the CIFAR-10 *vs* (CIFAR-100 or SVHN) tasks in Figure 3. As expected, the perceptual distance is better than the $\ell_2$ distance when $t_i$ is not too small. Our proposed metric, $S_\Pi^{\mathrm{LPIPS}}$ shows the best performance in the intermediate time step and this even exceeds the performance of diffusion-model-based OOD detection methods [17, 18] in the CIFAR-10 *vs* CIFAR-100 task. Furthermore, the performance of $S_\pi^{\mathrm{LPIPS}}$ degrades in larger time step due to the blurry output of the diffusion model $D_{\boldsymbol{\theta}}$. On the other hand, $S_\Pi^{\ell_2}$ and $S_\pi^{\ell_2}$ consistently struggle to detect the CIFAR-100 dataset and even output smaller abnormality scores on the SVHN dataset.

## 3.2 Projection Regret

While the perceptual distance performs better than the $\ell_2$ distance, it may fail to capture the semantic change when background information is dominant. For example, we refer to Figure 1b for the failure case of $S_\Pi^{\mathrm{LPIPS}}$. While the projection of the OOD (*i.e.*, bridge) data shows in-distribution (*i.e.*, church) semantics, the area of the changed region is relatively small. Hence, $S_\Pi^{\mathrm{LPIPS}}$ outputs a lower detection score for the OOD data than the in-distribution data.

To reduce such a background bias, we propose *Projection Regret*, a novel and effective detection method using *recursive projections* inspired by the observations in Section 3.1: (i) a projected image $\mathbf{x}' = \Pi_{\boldsymbol{\theta}}^{t_\alpha}(\mathbf{x})$ can be considered as an *in-distribution* image with a similar background to the original image $\mathbf{x}$, and (ii) its recursive projection $\Pi_{\boldsymbol{\theta}}^{t_\beta}(\mathbf{x}')$ may not change semantic information much because of (i). Therefore, the change in background statistics from $\mathbf{x}$ to $\Pi_{\boldsymbol{\theta}}^{t_\beta}(\mathbf{x})$ would be similar to that from $\mathbf{x}'$ to $\Pi_{\boldsymbol{\theta}}^{t_\beta}(\mathbf{x}')$, in other words, $d(\mathbf{x}, \Pi_{\boldsymbol{\theta}}^{t_\beta}(\mathbf{x})) \approx$ semantic difference $+ d(\mathbf{x}', \Pi_{\boldsymbol{\theta}}^{t_\beta}(\mathbf{x}'))$.

Based on this insight, we design our Projection Regret score, $S_{\mathrm{PR}}$, as follows:

$$S_{\mathrm{PR}}(\mathbf{x}, \alpha, \beta; \boldsymbol{\theta}) = S_\Pi^d(\mathbf{x}, t_\beta; \boldsymbol{\theta}) - \mathbb{E}_{\mathbf{z} \sim \mathcal{N}(\mathbf{0}, \mathbf{I})} \left[ S_\Pi^d(\Pi_{\boldsymbol{\theta}}^{t_\alpha}(\mathbf{x}, \mathbf{z}), t_\beta; \boldsymbol{\theta}) \right], \qquad (4)$$

where $\alpha$ and $\beta$ are timestep hyperparameters. We mainly use LPIPS [20] for the distance metric $d$. It is worth noting that Projection Regret is a plug-and-play module: any pre-trained diffusion model $\boldsymbol{\theta}$ and any distance metric $d$ can be used without extra training and information. The overall framework is illustrated in Figure 1b and this can be efficiently implemented by the consistency model $f_{\boldsymbol{\theta}}$ and batch-wise computation as described in Algorithm 1.

**Ensemble.** It is always challenging to select the best hyperparameters for the OOD detection task since we do not know the OOD information in advance. Furthermore, they may vary across different OOD datasets. To tackle this issue, we also suggest using an ensemble approach with multiple timestep pairs. Specifically, given a candidate set of pairs $\mathcal{C} = \{(\alpha_i, \beta_i)\}_{i=1}^{|\mathcal{C}|}$, we use the sum of all scores, *i.e.*, $S_{\mathrm{PR}}(\mathbf{x}; \boldsymbol{\theta}) = \sum_{(\alpha, \beta) \in \mathcal{C}} S_{\mathrm{PR}}(\mathbf{x}, \alpha, \beta; \boldsymbol{\theta})$, in our main experiments (Section 4.2) with a roughly selected $\mathcal{C}$.[1] We discuss the effect of each component in Section 4.3.

## 4 Experiment

In this section, we evaluate the out-of-distribution (OOD) detection performance of our Projection Regret. We first describe our experimental setups, including benchmarks, baselines, and implementation details of Projection Regret (Section 4.1). We then present our main results on the benchmarks (Section 4.2) and finally show ablation experiments of our design choices (Section 4.3).

## 4.1 Experimental setups

We follow the standard OOD detection setup [33, 34, 35]. We provide further details in the Appendix.

**Datasets.** We examine Projection Regret under extensive in-distribution (ID) *vs* out-of-distribution (OOD) evaluation tasks. We use CIFAR-10/100 [30] and SVHN [31] for ID datasets, and CIFAR-10/100, SVHN, LSUN [36], ImageNet [37], Textures [38], and Interpolated CIFAR-10 [19] for OOD datasets. Note that CIFAR-10, CIFAR-100, LSUN, and ImageNet have similar background information [39], so it is challenging to differentiate one from the other (*e.g.*, CIFAR-10 *vs* ImageNet). Following Tack et al. [39], we remove the artificial noise present in the resized LSUN and ImageNet

---

[1]We first search the best hyperparameter that detects rotated in-distribution data as OODs [32]. Then, we set $\mathcal{C}$ around the found hyperparameter.

Table 1: Out-of-distribution detection performance (AUROC) under various in-distribution *vs* out-of-distribution tasks. **Bold** and underline denotes the best and second best methods.

| Method | C10 *vs* | | | | C100 *vs* | | | SVHN *vs* | |
|---|---|---|---|---|---|---|---|---|---|
| | SVHN | CIFAR100 | LSUN | ImageNet | SVHN | C10 | LSUN | C10 | C100 |
| *Diffusion Models [26, 28]* | | | | | | | | | |
| Input Likelihood [6] | 0.180 | 0.520 | - | - | 0.193 | 0.495 | - | 0.974 | 0.970 |
| Input Complexity [7] | 0.870 | 0.568 | - | - | 0.792 | 0.468 | - | 0.973 | 0.976 |
| Likelihood Regret [8] | 0.904 | 0.546 | - | - | 0.896 | 0.484 | - | 0.805 | 0.821 |
| MSMA [17] | 0.992 | 0.579 | 0.587 | 0.716 | 0.974 | 0.426 | 0.400 | 0.976 | 0.979 |
| LMD [18] | 0.992 | 0.607 | - | - | **0.985** | 0.568 | - | 0.914 | 0.876 |
| *Consistency Models [21]* | | | | | | | | | |
| MSMA [17] | 0.707 | 0.570 | 0.605 | 0.578 | 0.643 | 0.506 | 0.559 | 0.985 | 0.981 |
| LMD [18] | 0.979 | 0.620 | 0.734 | 0.686 | 0.968 | 0.573 | 0.678 | 0.832 | 0.792 |
| Projection Regret (ours) | **0.993** | **0.775** | **0.837** | **0.814** | 0.945 | **0.577** | **0.682** | **0.995** | **0.993** |

Table 2: Out-of-distribution detection performance (AUROC) of Projection Regret against various generative models. **Bold** and underline denotes the best and second best methods.

| Method | Category | SVHN | CIFAR100 | CF10 Interp. | Textures | Avg Rank. |
|---|---|---|---|---|---|---|
| PixelCNN++ [48] | Autoregressive | 0.32 | 0.63 | 0.71 | 0.33 | 4.25 |
| GLOW [42] | Flow-based | 0.24 | 0.55 | 0.51 | 0.27 | 7.25 |
| NVAE [25] | VAE | 0.42 | 0.56 | 0.64 | - | 6.33 |
| IGEBM [19] | EBM | 0.63 | 0.50 | 0.70 | 0.48 | 5.25 |
| VAEBM [23] | VAE + EBM | 0.83 | 0.62 | 0.70 | - | 4.33 |
| Improved CD [22] | EBM | 0.91 | **0.83** | 0.65 | 0.88 | 3.25 |
| CLEL [24] | EBM | 0.98 | 0.71 | 0.72 | **0.94** | 2 |
| Projection Regret (ours) | Diffusion model | **0.99** | 0.78 | **0.80** | 0.91 | **1.5** |

datasets. We also remove overlapping categories from the OOD datasets for each task. We also apply Projection Regret in a one-class classification benchmark on the CIFAR-10 dataset. In this benchmark, we construct 10 subtasks where each single class in the corresponding subtask are chosen as in-distribution data and rest of the classes are selected as outlier distribution data. Finally, we also test Projection Regret in ColoredMNIST [40] benchmark where classifiers are biased towards spurious background information. All images are 32×32.

**Baselines.** We compare Projection Regret against various generative-model-based OOD detection methods. For the diffusion-model-based baselines, we compare Projection Regret against MSMA [17] and LMD [18]. We refer to Section 5 for further explanation. We implement MSMA[2] in the original NCSN and implement LMD in the consistency model. Furthermore, we experiment with the ablation of MSMA that compares $\ell_2$ distance between the input and its full-step projection instead of the single-step projection. Since likelihood evaluation is available in Score-SDE [28], we also compare Projection Regret with input likelihood [6], input complexity [7], and likelihood-regret [8] from the results of [18]. Finally, we compare Projection Regret with OOD detection methods that are based on alternative generative models, including VAE [41], GLOW [42], and energy-based models (EBMs) [19, 23, 22, 24]. Finally, for one-class classification, various reconstruction and generative model-based methods are included for reference (*e.g.*[43, 44, 45, 46]).

**Implementation details.** Consistency models are trained by consistency distillation in CIFAR-10/CIFAR-100 datasets and consistency training in the SVHN/ColoredMNIST datasets. We set the hyperparameters consistent with the author's choice [21] on CIFAR-10. We implement the code on the PyTorch [47] framework. All results are averaged across 3 random seeds. We refer to the Appendix for further details.

---

[2]We use the author's code in `https://github.com/ahsanMah/msma`.

Table 3: **One-class classification results (AUROC) of Projection Regret (PR) against various baselines in the CIFAR-10 one-class classification task. Bold** denotes the best method.

| Method | Category | Plane | Car | Bird | Cat | Deer | Dog | Frog | Horse | Ship | Truck | Mean |
|---|---|---|---|---|---|---|---|---|---|---|---|---|
| MemAE [43] | Autoencoder | - | - | - | - | - | - | - | - | - | - | 0.609 |
| AnoGAN [44] | GAN | 0.671 | 0.547 | 0.529 | 0.545 | 0.651 | 0.603 | 0.585 | 0.625 | 0.758 | 0.665 | 0.618 |
| OCGAN [45] | GAN | 0.757 | 0.531 | 0.640 | 0.620 | 0.723 | 0.620 | 0.723 | 0.575 | 0.820 | 0.554 | 0.657 |
| DROCC [46] | Adv. Generation | 0.817 | 0.767 | 0.667 | 0.671 | 0.736 | 0.744 | 0.744 | 0.714 | 0.800 | 0.762 | 0.742 |
| P-KDGAN [49] | GAN | 0.825 | 0.744 | 0.703 | 0.605 | 0.765 | 0.652 | 0.797 | 0.723 | 0.827 | 0.735 | 0.738 |
| LMD [18] | Diffusion Model | 0.770 | 0.801 | 0.719 | 0.506 | 0.790 | 0.625 | 0.793 | 0.766 | 0.740 | 0.760 | 0.727 |
| Rot+Trans [50] | Self-supervised | 0.760 | 0.961 | 0.867 | 0.772 | 0.911 | 0.885 | 0.884 | 0.961 | 0.943 | 0.918 | 0.886 |
| Projection Regret **(ours)** | Diffusion Model | **0.864** | 0.946 | 0.789 | 0.697 | 0.840 | 0.825 | 0.854 | 0.889 | 0.929 | 0.919 | 0.855 |
| PR + (Rot+Trans) **(ours)** | Hybrid | 0.829 | **0.971** | **0.889** | **0.803** | **0.925** | **0.905** | **0.911** | **0.963** | **0.954** | **0.933** | **0.908** |

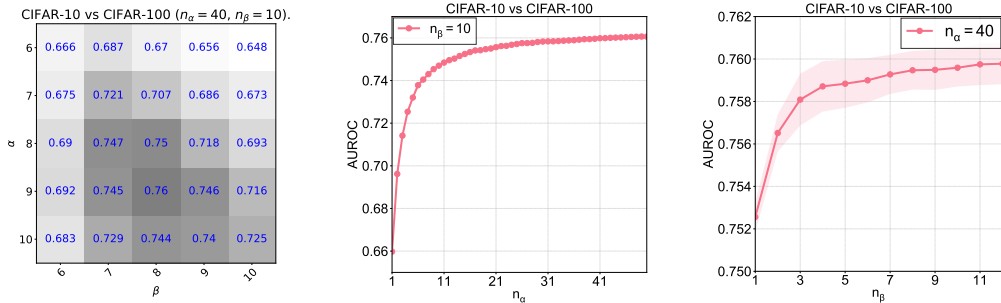

Figure 4: **Hyperparameter analysis of Projection Regret on CIFAR-10 vs CIFAR-100 task. (left):** Ablation on hyperparameter $\alpha$ and $\beta$. The proposed time ensemble performs better than the best hyperparameter. **(middle):** Ablation on hyperparameter $n_\alpha$. **(right):** Ablation on hyperparameter $n_\beta$.

## 4.2 Main results

Table 1 shows the OOD-detection performance against diffusion-model-based methods. Projection Regret consistently outperforms MSMA and LMD on 8 out of 9 benchmarks. While Projection Regret mainly focuses on OODs with similar backgrounds, Projection Regret also outperforms baseline methods on the task where they show reasonable performance (*e.g.*, CIFAR-10 *vs* SVHN).

In contrast, MSMA struggles on various benchmarks where the ID and the OOD share similar backgrounds (*e.g.*, CIFAR-10 *vs* CIFAR-100). While LMD performs better in such benchmarks, LMD shows underwhelming performance in SVHN *vs* (CIFAR-10 or CIFAR-100) and even underperforms over input likelihood [6]. To analyze this phenomenon, we perform a visualization of the SVHN samples that show the highest abnormality score and their inpainting reconstructions in the Appendix. The masking strategy of LMD sometimes discards semantics and thereby reconstructs a different image. In contrast, Projection Regret shows near-perfect OOD detection performance in these tasks.

We further compare Projection Regret against OOD detection methods based on alternative generative models (*e.g.*, EBM [24], GLOW [42]) in Table 2. We also denote the category of each baseline method. Among the various generative-model-based OOD detection methods, Projection Regret shows the best performance on average. Significantly, while CLEL [24] requires an extra self-supervised model for joint modeling, Projection Regret outperforms CLEL in 3 out of 4 tasks.

We finally compare Projection Regret against various one-class classification methods in Table 3. Projection Regret achieves the best performance and outperforms other reconstruction-based methods consistently. Furthermore, when combined with the contrastive learning method [50] by simply multiplying the anomaly score, our approach consistently improves the performance of the contrastive training-based learning method. Specifically, Projection Regret outperforms the contrastive training-based method in airplane class where the data contains images captured in varying degrees. As such, contrastive training-based methods underperform since they use rotation-based augmentation to generate pseudo-outlier data.

## 4.3 Analysis

**Hyperparameter analysis.** We analyze the effect of hyperparameters for Projection Regret under the CIFAR-10 *vs* CIFAR-100 task. First, we conduct experiments with varying the timestep index

Table 4: **Ablation Study on each component of Projection Regret.** In-distribution dataset is the CIFAR-10 dataset. we report AUROC. Gains are computed against LMD [18].

| Method | SVHN | CIFAR100 | LSUN | ImageNet |
|---|---|---|---|---|
| LMD [18] | 0.979 | 0.620 | 0.734 | 0.686 |
| Projection (*i.e.*, $S_{\Pi}^{\text{LPIPS}}$) | 0.966 (-1.3%) | 0.667 (+4.7%) | 0.762 (+2.8%) | 0.718 (+3.2%) |
| Projection Regret ($\alpha = 9, \beta = 8$) | 0.986 (+0.7%) | 0.760 (+14.0%) | 0.828 (+9.4%) | 0.795 (+10.9%) |
| Projection Regret (Ensemble) | 0.993 (+1.4%) | 0.775 (+15.5%) | 0.837 (+10.3%) | 0.814 (+12.8%) |

Table 5: AUROC of Projection Regret given different distance metric choices. **Bold** and underline denotes the best and second best methods.

| Method | Distance $d$ | SVHN | CIFAR100 | LSUN | ImageNet |
|---|---|---|---|---|---|
| LMD [18] | | 0.979 | 0.620 | 0.734 | 0.686 |
| Projection Regret | SSIM [27] | 0.328 | 0.629 | 0.650 | 0.620 |
| | LPIPS [20] | **0.993** | **0.775** | 0.837 | 0.814 |
| | UNet (proposed) | 0.917 | 0.734 | **0.865** | **0.815** |

hyperparameters $\alpha$ and $\beta$. As mentioned in Section 3.2, the ensemble performs better (*i.e.*, 0.775 in Table 1) than the best hyperparameter selected (*i.e.*, 0.760 when $\alpha = 9, \beta = 8$).

Furthermore, we perform ablation on the ensemble size hyperparameter $n_\alpha$ and $n_\beta$. While selecting a larger ensemble size improves the performance, we show reasonable performance can be achieved in smaller $n_\alpha$ or $n_\beta$ (*e.g.*, $n_\alpha = 40, n_\beta = 5$ or $n_\alpha = 20, n_\beta = 10$) without much sacrifice on the performance.

**Ablation on the components of Projection Regret.** We further ablate the contribution of each component in Projection Regret under the CIFAR-10 dataset. First, we set the best-performing timestep index hyperparameter ($\alpha = 9, \beta = 8$) under the CIFAR-10 *vs* CIFAR-100 detection task. Furthermore, we also compare the projection baseline method, *i.e.*, $S_{\Pi}^{\text{LPIPS}}(\cdot, t_\beta; \boldsymbol{\theta})$ where $\beta = 8$.

The performance of each method is shown in Table 4. We also denote the absolute improvements over LMD [18]. Our proposed baseline method $S_{\Pi}^{\text{LPIPS}}$ already improves over LMD in 3 out of 4 tasks, especially in the CIFAR-10 OOD dataset with similar background statistics. Furthermore, we observe consistent gains on both Projection Regret with and without ensemble over baselines. The largest gain is observed in Projection Regret over the projection baseline $S_{\Pi}^{\text{LPIPS}}$ (*e.g.*, 9.3% in the CIFAR-10 *vs* CIFAR-100 detection task).

**Diffusion-model-based distance metric.** While we mainly use LPIPS [20] as the distance metric $d$, we here explore alternative choices on the distance metric.

We first explore the metric that can be extracted directly from the pre-trained diffusion model. To this end, we consider intermediate feature maps of the U-Net [51] decoder of the consistency model $f_{\boldsymbol{\theta}}$ since the feature maps may contain semantic information to reconstruct the original image [52]. Formally, let $F^{(\ell)}(\mathbf{x}, t, \mathbf{z}) \in \mathbb{R}^{C_\ell \times H_\ell \times W_\ell}$ be the $\ell$-th decoder feature map of $f_{\boldsymbol{\theta}}(\mathbf{x} + t\mathbf{z}, t)$. Motivated by semantic segmentation literature [53, 54] that utilize the cosine distance between two feature maps, we define our U-Net-based distance metric $d_{\texttt{UNet}}(\cdot, \cdot)$ as follows:

$$d_{\texttt{UNet}}(\mathbf{x}, \mathbf{x}') = \mathbb{E}_{\mathbf{z} \sim \mathcal{N}(\mathbf{0}, \mathbf{I})} \Big[ \sum_{\ell=1}^{L} \frac{1}{H_\ell W_\ell} \sum_{ij} d_{\texttt{cos}} \Big( F^{(\ell)}_{:,i,j}(\mathbf{x}, t_\gamma, \mathbf{z}), F^{(\ell)}_{:,i,j}(\mathbf{x}', t_\gamma, \mathbf{z}) \Big) \Big], \quad (5)$$

where $d_{\texttt{cos}}(\mathbf{u}, \mathbf{v}) = \big\| \mathbf{u}/\|\mathbf{u}\|_2 - \mathbf{v}/\|\mathbf{v}\|_2 \big\|_2^2$ is the cosine distance and $t_\gamma$ is a timestep hyperparameter. We set $t_\gamma$ small enough to reconstruct the original image, *e.g.*, $t_\gamma = 0.06$ (*i.e.*, $\gamma = 3$), for evaluation.

We report the result in Table 5. We also test the negative SSIM index [27] as the baseline distance metric. While LPIPS performs the best, our proposed distance $d_{\texttt{UNet}}$ shows competitive performance against LPIPS and outperforms LMD and MSMA in CIFAR-10 *vs* (CIFAR-100, LSUN or ImageNet) tasks. On the other hand, SSIM harshly underperforms LPIPS and $d_{\texttt{UNet}}$.

**Using the diffusion model for full-step projection.** While the consistency model is used for full-step projection throughout our experiments, a diffusion model can also generate full-step projection $\Pi_{\boldsymbol{\theta}}^t(\mathbf{x})$

Table 6: Performance (AUROC) of Projection Regret computed in the EDM [29] with Heun's 2nd order sampler. Performance of Projection Regret under consistency model [21] under the same hyperparameter is included for comparison.

| Model | SVHN | CIFAR100 | LSUN | ImageNet |
|---|---|---|---|---|
| $n_\alpha = 10, n_\beta = 5$ | | | | |
| EDM [29] | 0.986 | 0.763 | 0.825 | 0.799 |
| CM [21] | 0.990 | 0.761 | 0.816 | 0.797 |
| $n_\alpha = 40, n_\beta = 10$ | | | | |
| CM [21] | 0.993 | 0.775 | 0.837 | 0.814 |

Table 7: **OOD detection results (TNR at 95% TPR, AUROC) under more severe background bias.** In-distribution dataset is the colored MNIST dataset in Ming et al. [55] with $r = 0.45$. The result in Ming et al. [55] is included for reference. Higher is better in all metrics.

| Method | Spurious OOD dataset | LSUN | iSUN | Textures |
|---|---|---|---|---|
| Ming et al. [55] | 0.696/0.867 | 0.901/0.959 | 0.933/0.982 | 0.937/0.985 |
| Projection Regret | 0.958/0.990 | 1.000/1.000 | 1.000/1.000 | 1.000/1.000 |

with better generation quality through multi-step sampling. For verification, we experiment with the EDM [29] trained in the CIFAR-10 dataset and perform full-step projection via the 2nd-order Heun's sampler. Since sampling by diffusion model increases the computation cost by $2\beta$ times compared to the consistency model, we instead experiment on the smaller number of ensemble size $n_\alpha = 10, n_\beta = 5$ and report the result of Projection Regret with a consistency model computed in the same hyperparameter setup as a baseline.

We report the result in Table 6. We denote the result of Projection Regret utilizing the full-step projection based on the diffusion model and the consistency model as EDM and CM, respectively. Despite its excessive computation cost, Projection Regret on EDM shows negligible performance gains compared to that on CM.

**Projection Regret helps robustness against spurious correlations.** While we mainly discuss the problem of generative model overfitting into background information in the unsupervised domain, such a phenomenon is also observed in supervised learning where the classifier overfits the background or texture information [56]. Motivated by the results, we further evaluate our results against the spurious OODs where the OOD data share a similar background to the ID data but with different semantics. Specifically, we experiment on the coloredMNIST [40] dataset following the protocol of [55]. We report the result in Table 7. Note that spurious OOD data have the same background statistics and different semantic (digit) information. Projection Regret achieves near-perfect OOD detection performance. It is worth noting that Projection Regret does not use any additional information while the baseline proposed by [55] uses label information.

## 5 Related Works

**Fast sampling of diffusion models.** While we use full-step projection of the image for Projection Regret, this induces excessive sampling complexity in standard diffusion models [26, 57, 28]. Hence, we discuss works that efficiently reduce the sampling complexity of the diffusion models. PNDM [58] and GENIE [59] utilize multi-step solvers to reduce the approximation error when sampling with low NFEs. Salimans and Ho [60] iteratively distill the 2-step DDIM [61] sampler of a teacher network into a student network.

**Out-of-distribution detection on generative models.** Generative models have been widely applied to out-of-distribution detection since they implicitly model the in-distribution data distribution. A straightforward application is directly using the likelihood of the explicit likelihood model (*e.g.*, VAE [41]) as a normality score function [6]. However, Nalisnick et al. [33] discover that such models may assign a higher likelihood to OOD data. A plethora of research has been devoted to explaining the aforementioned phenomenon [62, 7, 63, 64]. Serrà et al. [7] hypothesize that deep generative models assign higher likelihood on the simple out-of-distribution dataset. Kirichenko et al. [63] propose that flow-based models' [42] likelihood is dominated by local pixel correlations.

Finally, diffusion models have been applied for OOD detection due to their superior generation capability. Mahmood et al. [17] estimate the $\ell_2$ distance between the input image and its single-step projection as the abnormality score function. Liu et al. [18] applies checkerboard masking to the input image and performs a diffusion-model-based inpainting task. Then, reconstruction loss between the inpainted image and the original image is set as the abnormality score function.

**Reconstruction-based out-of-distribution detection methods.** Instead of directly applying pre-trained generative models for novelty detection, several methods[43, 46, 45] train a reconstruction model for out-of-distribution. MemAE [43] utilizes a memory structure that stores normal representation. DROCC [46] interprets one-class classification as binary classification and generates boundary data via adversarial generation. Similarly, OCGAN [45] trains an auxiliary classifier that controls the generative latent space of the GAN model. While these methods are mainly proposed for one-class classification, we are able to outperform them with significant gains.

**Debiasing classifiers against spurious correlations.** Our objective to capture semantic information for novelty detection aligns with lines of research that aim to debias classifiers against spurious correlations [40, 56, 65, 66, 67]. Geirhos et al. [56] first observe that CNN-trained classifiers show vulnerability under the shift of non-semantic regions. Izmailov et al. [66] tackles the issue by group robustness methods. Motivated by research on interpretability, Wu et al. [67] apply Grad-CAM [68] to discover semantic features. Ming et al. [55] extend such a line of research to out-of-distribution detection by testing OOD detection algorithms on spurious anomalies.

# 6    Conclusion

In this paper, we propose Projection Regret, a novel and effective out-of-distribution (OOD) detection method based on the diffusion-based projection that transforms any input image into an in-distribution image with similar background statistics. We also utilize recursive projections to reduce background bias for improving OOD detection. Extensive experiments demonstrate the efficacy of Projection Regret on various benchmarks against the state-of-the-art generative-model-based methods. We hope that our work will guide new interesting research directions in developing diffusion-based methods for not only OOD detection but also other downstream tasks such as image-to-image translation [69].

**Limitation.** Throughout our experiments, we mainly use the consistency models to compute the full-step projection via one network evaluation. This efficiency is a double-edged sword, *e.g.*, consistency models often generate low-quality images compared to diffusion models. This might degrade the detection accuracy of our Projection Regret, especially when the training distribution $p_{\text{data}}$ is difficult to learn, *e.g.*, high-resolution images. However, as we observed in Table 6, we strongly believe that our approach can provide a strong signal to detect out-of-distribution samples even if the consistency model is imperfect. In addition, we also think there is room for improving the detection performance by utilizing internal behaviors of the model like our distance metric $d_{\text{UNet}}$. Furthermore, since diffusion models use computation-heavy U-Net architecture, our computational speed is much slower than other reconstruction-based methods.

**Negative social impact.** While near-perfect novelty detection algorithms are crucial for real-life applications, they can be misused against societal agreement. For example, an unauthorized data scraping model may utilize the novelty detection algorithm to effectively collect the widespread data. Furthermore, the novelty detection method can be misused to keep surveillance on minority groups.

## Acknowledgments and Disclosure of Funding

This work is fully supported by LG AI Research.

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

# Supplementary Material

## Projection Regret: Reducing Background Bias
## for Novelty Detection via Diffusion Models

## A  Full algorithm

We refer to Algorithm 1 for the efficient computation of our method.

---

**Algorithm 1** Projection Regret with Consistency Models (PyTorch-like Pseudo-code)

---

```
# f(x, t): consistency model
# t_alpha, t_beta: the projection timesteps
# n_alpha, n_beta: the projection ensemble sizes
# repeat(A, n): repeat tensor A n times along axis=0
# d(a, b): distance function, e.g., LPIPS

def Projection(x, t):
    return f(x + t * randn_like(x), t)

def ProjectionRegret(x): # x: [1, 3, H, W]
    x_proj = Projection(repeat(x, n_alpha * n_beta),  t_beta)
    y      = Projection(repeat(x, n_alpha), t_alpha)
    y_proj = Projection(repeat(y, n_beta),  t_beta)
    dx = d(repeat(x, n_alpha * n_beta), x_proj).mean()
    dy = d(repeat(y, n_beta), y_proj).mean()
    return dx - dy
```

---

## B  Implementation details

We train the baseline EDM [29] model for 390k iterations for all datasets. Then, we perform consistency distillation [21] for the CIFAR-10 and CIFAR-100 [30] datasets and consistency training for the SVHN [31] dataset. The resulting FID score [70] of the CIFAR-10/CIFAR-100 dataset is 5.52/7.0 and 8.0 for the SVHN dataset (the model trained by consistency training usually shows higher FID than the one trained by consistency distillation). While training the base EDM diffusion model and the consistency model, we use 8 A100 GPUs. During the out-of-distribution detection, we use 1 A100 GPU. For the ensemble size hyperparameter $n_\alpha$ and $n_\beta$, we set $n_\alpha = 40, n_\beta = 10$ for CIFAR-10/SVHN experiments and $n_\alpha = 100, n_\beta = 5$ for CIFAR-100 experiment. As mentioned in the main paper, we set our base hyperparameter that shows the best performance against the rotated in-distribution dataset and set the ensemble configurations around it. We use 8 hyperparameter configurations for the CIFAR-10 and CIFAR-100 datasets and 4 hyperparameter configurations for the SVHN dataset. To be specific, chosen configurations are $\mathcal{C} = \{(7,6),(7,7),(8,7),(8,8),(9,8),(9,9),(10,9),(10,10)\}$ for CIFAR-10/CIFAR-100 dataset and $\mathcal{C} = \{(10,10),(11,10),(11,11),(12,11)\}$ for the SVHN dataset.

## C  Visualization of LMD in SVHN dataset

We further reconstruct the output of the LMD [18] when SVHN is the in-distribution dataset. Following the practice of Liu et al. [18], we set the mask as the alternating checkboard mask throughout the experiments in Section 4.2. We visualize a grid of 9 samples and their reconstructions where LMD shows the highest abnormality score in Figure 5. We can see LMD's inconsistent reconstruction in such data, thereby failing to reconstruct consistent images. Furthermore, as mentioned in Section 4.2, sometimes the checkboard strategy discards the in-distribution data's semantics and thereby fails to reconstruct the original image.

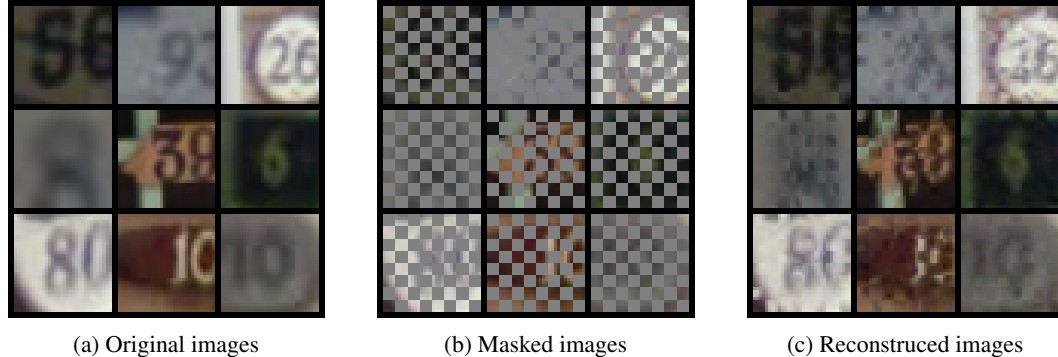

(a) Original images      (b) Masked images      (c) Reconstruced images

Figure 5: **Visualization of LMD [18] in the SVHN [31] dataset. (a)** Original SVHN images where LMD outputs the highest abnormality score. **(b)** The masked images following the practice of Liu et al. [18]. **(c)** Their reconstruction images through LMD.

