# OpenReview forum: "Projection Regret: Reducing Background Bias for Novelty Detection via Diffusion Models"
_NeurIPS.cc/2023/Conference — NeurIPS 2023 poster_

### Official Review · Reviewer_SZQH · 2023-06-30

**Soundness:** 2 fair
**Presentation:** 3 good
**Contribution:** 2 fair
**Rating:** 5
**Confidence:** 4

**Summary:**

Recent methods have mainly utilized the reconstruction property of in-distribution samples to detect OOD by diffusion models. However, they often suffer from detecting OOD samples that share similar background information to the in-distribution data. Based on the observation that diffusion models can project any sample to an in-distribution sample with similar background information, the paper proposes Projection Regret (PR), an efficient novelty detection method that mitigates the bias of non-semantic information. To be specific, PR computes the perceptual distance between the test image and its recursive diffusion-based projection to detect the abnormality.

**Strengths:**

1. The paper is written well and is easy to understand.
2. The studied problem is very important.
3. The results seem to outperform state-of-the-art.

**Weaknesses:**

1. From the introduction, the projection cannot change the background information too much, which seems to be harmful on near-OOD detection, such as Cifar10 vs Cifar100. Since c100 and c10 has a similar background, then wouldn't this projection create a barrier further for solving the OOD detection problem?
2. The design and hyperparameter search (including the ensemble) seems to be very important on the results. How do the authors get the best configuration on a new ID/OOD pair? Is there any sensitivity analysis on more pairs? i.e., doe the current Figure 3 generalize to most ID/OOD pairs?
3. The computational budget is a little bit concerning in the algorithm. Is there any concrete comparison or explanation?
4. More results on large-scale benchmarks (i.e., ImageNet) and large (diffusion) models seem to be more meaningful for practical concerns and interpretability.

**Questions:**

see above

**Limitations:**

yes

---

> ### Author Rebuttal · Authors · 2023-08-09
>
> Dear Reviewer SZQH,
>
> We sincerely thank you for your helpful feedback and insightful comments. In what follows, we address your concerns one by one.
> ___
> **[Q1]** From the introduction, the projection cannot change the background information too much, which seems to be harmful on near-OOD detection, such as Cifar10 vs Cifar100. Since c100 and c10 has a similar background, then wouldn't this projection create a barrier further to solving the OOD detection problem?\
> **[A1]** We clarify that our projection of an out-of-distribution (OOD) sample changes its semantics from OOD to in-distribution (see Figure 1(a)). Since our detection score is based on the semantic distance metric between a test image and its projection, our algorithm will output a high abnormality score on the OOD dataset even if the background information is similar.
>
> This claim is evidenced by our empirical results: our algorithm significantly outperforms baselines on CIFAR-10 vs CIFAR-100 (see Table 1). To further address your concern, we additionally experiment with an additional OOD benchmark, the ColorMNIST dataset [1,2] where spurious OOD samples have **the same background statistics** (i.e., same background colors) but different semantics (i.e., different digits) compared to the in-distribution data. In this dataset, our algorithm achieves near-perfect OOD detection performance as shown in Table 6 (see Global Response PDF). These experimental results verify that our method can detect OOD samples that have similar backgrounds to the in-distribution samples.
> ___
> **[Q2]** The design and hyperparameter search (including the ensemble) seems to be very important in the results. How do the authors get the best configuration on a new ID/OOD pair? Is there any sensitivity analysis on more pairs? i.e., doe the current Figure 3 generalize to most ID/OOD pairs?\
> **[A2]** We choose the hyperparameters for each in-distribution (ID) data without any access to out-of-distribution (OOD) samples as described in footnote 1 on page 5. To be specific, we consider rotated in-distribution samples as synthetic OOD ones and choose the best hyperparameters in this synthetic OOD detection task. We find that the hyperparameters selected from the synthetic task are well generalized across other OOD datasets, as we verified in our experiments. Therefore, for a new ID dataset, one can find the best configuration itself, and the configuration can be used across various OOD pairs.
>
> For your information, we further provide the sensitivity analysis of Projection Regret on more ID/OOD pairs: CIFAR-10 vs (SVHN, LSUN, ImageNet) datasets in Figure 6 (see Global Response PDF). Somewhat interestingly, the hyperparameter chosen by our synthetic OOD detection task ($\alpha$=9, $\beta$=8) also achieves the best performance on LSUN and ImageNet datasets or a small gap (0.005 AUROC) against the best performance on the SVHN dataset. We further improve this by ensembling across multiple configurations and the ensemble outperforms the best-searched hyperparameter (see Table 3).
>
> Finally, as the reviewer asked about the generalization property of Figure 3 to other ID/OOD pairs, we also test our projection with varying timesteps. As shown in Figure 6 (see Global Response PDF), the trend in the LSUN and ImageNet OOD datasets shows a similar trend to CIFAR-100 and SVHN shown in Figure 3. Therefore, we think the hyperparameters are not sensitive to ID/OOD pairs, which is a merit of our framework.
> ___
> **[Q3]** The computational budget is a little bit concerning in the algorithm. Is there any concrete comparison or explanation?\
> **[A3]** Our algorithm is efficient because it only requires three projections and each projection can be computed by one forward pass using a consistency model [3] that enables one-shot generation, as described in Algorithm 1. As a result, even with our ensemble technique described in L162-L167, our algorithm is 1.8x faster than the second-best baseline, LMD [4] (0.395s vs 0.705s per sample, respectively).
> ___
> **[Q4]** More results on large-scale benchmarks (i.e., ImageNet) and large (diffusion) models seem to be more meaningful for practical concerns and interpretability.\
> **[A4]** To show our algorithm’s scalability in large-scale datasets with high-resolution images, we construct an OOD detection task on the LSUN domain. Specifically, we set the bedroom class of LSUN as an in-distribution dataset and apply the pre-trained consistency model to detect bridge/church/classroom OOD datasets. We also experiment with LMD as a baseline. As reported in Table 8 (see Global Response PDF), our method significantly outperforms LMD by a large margin across all large-scale LSUN OOD detection tasks. We also observe that our method achieves better AUROC in relatively far OOD datasets (bridge, church) against near OOD datasets (classroom). Hence, this additional experimental result shows the scalability and the potential of practical applications.
> ___
>
> **References** \
> [1] Invariant risk minimization, arXiv 2019\
> [2] On the impact of spurious correlation for out-of-distribution detection, AAAI 2022\
> [3] Consistency Models, ICML 2023\
> [4] Unsupervised out-of-distribution detection with diffusion inpainting, ICML 2023

---

> > ### Comment · Reviewer_SZQH · 2023-08-17
> > **Acknowledgement**
> >
> > Thanks for your response to address my questions. I have increased the score to 5.

---

> > > ### Author Response · Authors · 2023-08-17
> > >
> > > We are glad that our responses addressed your concerns. We will incorporate the additional results in the final manuscript.
> > >
> > >
> > > Thank you!
> > >
> > > Authors.

---

### Official Review · Reviewer_9KBe · 2023-07-02

**Soundness:** 2 fair
**Presentation:** 2 fair
**Contribution:** 2 fair
**Rating:** 5
**Confidence:** 2

**Summary:**

This paper discusses a method in machine learning known as Novelty Detection, which is used to identify abnormal or out-of-distribution (OOD) samples. The authors suggest that diffusion models, a popular generative framework due to their strong generation performance, has recently become an attractive tool for novelty detection.

However, the authors note a problem: while these models excel at generating high-quality results, they often struggle to detect OOD images with similar backgrounds. This issue, referred to as 'background bias', can lead to inaccurate novelty detection.

To address this issue, the paper proposes 'Projection Regret' (PR), a method which reduces the impact of background bias and improves the accuracy of novelty detection. PR calculates the perceptual distance between a test image and its 'projection' (an in-distribution sample with similar background information created by the diffusion model). To further mitigate the effect of dominant background information, PR uses recursive projections to cancel out the background bias.

The paper also introduces an ensemble of multiple projections for improved detection performance, calculated efficiently via a consistency model. A new perceptual distance metric using underlying features of the diffusion model is proposed and compared to other metrics, showing promising results.

In conclusion, the paper's main contributions are:
1. Identifying the issue of background bias in novelty detection via diffusion models.
2. Proposing a solution, Projection Regret (PR), which mitigates this bias and enhances OOD detection.
3. Introducing an alternative perceptual distance metric computed from decoder features of the pre-trained diffusion model.

Extensive experiments demonstrate that PR significantly outperforms previous diffusion-based novelty detection methods, showing potential for future applications.

**Strengths:**

**Originality:** The paper presents a novel method called Projection Regret (PR) to address the problem of background bias in novelty detection using diffusion models. This is a creative combination of existing ideas, especially the use of perceptual distance and recursive projections to cancel out dominant background information. The proposed alternative perceptual distance metric using decoder features of the pre-trained diffusion model also adds to the originality.

**Quality:** The research appears to be of high quality. The authors have carried out extensive experiments to demonstrate the effectiveness of their proposed method. They have also compared PR with other existing methods for novelty detection, showing that it outperforms them by a significant margin.

**Clarity:** The paper is well written and structured. The authors clearly outline the problem of background bias in novelty detection, explain the concept behind PR, and provide detailed explanations of their experimental setup and results. They also do a good job of explaining complex concepts in an understandable way.

**Significance:** The significance of this work lies in its potential to improve the accuracy of novelty detection in machine learning, which has broad applications in many fields, including medical diagnosis, autonomous driving, and forecasting. By addressing the issue of background bias, the proposed PR method could enhance the reliability and safety of deep learning applications. Additionally, the proposed perceptual distance metric could serve as a useful tool for researchers working on similar problems in the future.

**Weaknesses:**

While the paper presents a significant contribution to the field, there are a few areas where it could be improved:

**1. Evaluation metrics:** While the paper does an excellent job of comparing with existing methods and demonstrating the superiority of PR, it might benefit from more diverse evaluation metrics. At present, the paper primarily focuses on detection accuracy (AUROC). Incorporating additional measures such as precision-recall curves or F1 scores could provide a more comprehensive evaluation.

**2. Real-world applications:** The paper demonstrates PR's effectiveness using standard datasets (CIFAR-10 vs CIFAR-100 etc.), but it would be valuable to see how PR performs in real-world scenarios. It is crucial to know how the method copes with the complexity and variability found in actual application scenarios including medical imaging or self-driving car data.

**3. Deeper exploration of background bias:** Although the authors have proposed an innovative solution to tackle the issue of background bias, they could delve deeper into this problem. Understanding the nature of this bias, its origins, and why it is particularly problematic for diffusion models can further enhance the paper's impact.

**4. Computational efficiency:** While the authors mention that PR can be calculated efficiently via a consistency model, specific data about computational requirements, time complexity, and scalability is missing. Providing these details can help readers assess whether PR is suitable for their specific use-cases, especially those that require real-time processing or deal with large-scale datasets.

**5. Robustness analysis:** The paper could benefit from a detailed robustness analysis of PR against various types of noise and distortions. This would allow potential users to understand the limits of the approach and potential pitfalls in practical applications.

By addressing these points, the paper could strengthen its contributions and appeal to a broader audience within the machine learning community.

**Questions:**

**Questions:**

1. Could you provide more details on the computational requirements of PR? For instance, what is the time complexity and how does it scale with the size of the dataset?

2. Would PR be equally effective in real-world applications where data might be more complex or noisy compared to standard datasets like CIFAR-10 or CIFAR-100?

3. Could you delve deeper into the issue of background bias? Understanding its origins and why it affects diffusion models specifically could strengthen the paper.

4. Have you investigated the robustness of PR against various types of noise and distortions? If not, do you anticipate that the method would be robust against such challenges?

**Suggestions:**

1. Consider incorporating additional evaluation metrics such as precision-recall curves or F1 scores for a more comprehensive evaluation of PR's performance.

2. It would be beneficial to demonstrate how PR performs with real-world datasets, such as medical imaging data or self-driving car sensory data. This would help readers understand its applicability and effectiveness in practical scenarios.

3. Detailed information about computational efficiency, including time complexity and scalability aspects, would be useful for potential users assessing suitability for their specific use-cases.

4. Conducting a robustness analysis against various types of noise and distortions would further validate the efficacy of PR under different challenging conditions.



**Limitations:**

The authors have addressed some limitations of their work, particularly the issue of background bias in diffusion models and how their proposed method, Projection Regret (PR), mitigates this. However, a few additional limitations could be further discussed:

1. **Computational Resources:** While the authors mention that PR can be calculated efficiently, the paper does not provide specific details on computational requirements or scalability. As advanced machine learning models often require significant computational resources, an evaluation of this aspect would be helpful for potential users.

2. **Robustness:** The paper does not explicitly discuss the robustness of PR to noise or distortions in the data. Given that real-world data can often be noisy or imperfect, discussing the limitations of PR in handling such situations would be beneficial.

Regarding the broader societal impacts, the paper does not directly address this point. While novelty detection in machine learning can have multiple positive societal impacts, like enhancing medical diagnosis or improving safety in autonomous driving, it might also have potential negative impacts:

1. **Privacy concerns:** Improving the ability of machines to identify novel information could potentially lead to privacy concerns, as more accurate models might be misused for surveillance or unauthorized data collection.

2. **System Misuse:** In high-stakes applications, an over-reliance on automated novelty detection systems that might still make mistakes could lead to serious consequences.

The authors could improve their paper by addressing these points, possibly in a new section dedicated to limitations and broader societal impacts. This would help readers appreciate the full context of the research, including its potential downsides.

---

> ### Author Rebuttal · Authors · 2023-08-09
>
> Dear reviewer 9KBe,
>
> We sincerely thank you for your helpful feedback and insightful comments. In what follows, we address your concerns one by one.
> ___
> **[Q1]** Could you provide more details on the computational requirements of PR? For instance, what is the time complexity, and how does it scale with the size of the dataset?\
> **[A1]** As shown in Algorithm 1, our algorithm (Projection Regret or PR) performs 3 forward computations of the consistency model. Therefore, the time complexity is linear to the size of the dataset. In a real-time setting, our algorithm takes 0.395s/sample in the CIFAR-10 dataset and the consistency model version of LMD [1] takes 0.705s on the same dataset.
> ___
>  **[Q2]** Would PR be equally effective in real-world applications where data might be more complex or noisy compared to standard datasets like CIFAR-10 or CIFAR-100?\
> **[A2]** We scale Projection Regret into a large-scale LSUN domain where the dataset is more complex. To be specific, we set the bedroom class as the in-distribution (ID) dataset and set bridge/church/classroom classes as the out-of-distribution (OOD) dataset. We summarize the OOD detection result in Table 8 (see Global Response PDF). Our method outperforms the second-best method, LMD [1], with a significant gap. This result shows the potential of PR in real-world applications.
> ___
> **[Q3]** Could you delve deeper into the issue of background bias? Understanding its origins and why it affects diffusion models specifically could strengthen the paper.\
> **[A3]**  The background bias issue originates from the fact that we do not have a perfect distance metric that only captures the semantic difference between two images. While the LPIPS distance metric [2] mainly used in our work captures such a semantic change better than $\ell_{2}$ distance, it often captures other information (e.g., background), especially when the main object is relatively small (see L49-L50). Furthermore, when we apply the metric on various datasets that significantly differ from ImageNet, e.g., SVHN or LSUN, it is not likely that the LPIPS distance metric fully captures the semantic change between the input data and its projection. In such scenarios, the distance metric could be biased by the background information as shown in Figure 1(b). Hence, we think the background bias issue is problematic due to an imperfect distance metric.
>
> To reduce the background bias, we propose Projection Regret using the unique characteristic of diffusion models: diffusion-based projection can map any OOD sample to an ID one with a similar background. Therefore, we rather regard diffusion models (or consistency models) as the remedy to compensate for the imperfect distance metric. Furthermore, it is evident that OOD detection methods based on alternative generative models, e.g. GANs or GLOW, are also vulnerable to OOD with a similar background (see CIFAR-10 vs CIFAR-100 OOD detection results in Table 2). Hence, we think background bias is not limited to the diffusion model but a common problem of unsupervised OOD detection methods.
> ___
> **[Q4]** Have you investigated the robustness of PR against various types of noise and distortions? If not, do you anticipate that the method would be robust against such challenges?\
> **[A4]** We appreciate the reviewer for the motivating suggestion. We further test PR’s robustness under the corruption of the in-distribution data. To be specific, we set **corrupted** CIFAR-10 data [3] as pseudo-in-distribution data and perform an OOD detection task against **uncorrupted** SVHN/CIFAR-100/LSUN/ImageNet as OOD datasets. For the corruption strategy, we use (shot noise, defocus blur, fog, and elastic transform) for the representative strategy of (noise, blur, weather, and digital) in varying corruption levels of (1,2,3). We provide the visualization of the corrupted data in Figure 5 (see Global Response PDF).
>
> We present the result in Table 9 (see Global Response PDF). As corruption intensity increases, the OOD detection performance decreases. While strong corruptions (e.g. shot noise, fog) deteriorate OOD detection performance significantly, our method is relatively robust to weak corruption (e.g. defocus blur). Hence, our method would be at least robust to imperceptible corruption.
> ___
> **[Q5]** Consider incorporating additional evaluation metrics such as precision-recall curves or F1 scores for a more comprehensive evaluation of PR's performance.\
> **[A5]**  Widely used popular metrics on OOD detection (TNR at 95\% TPR, detection accuracy, AUPR_IN, and AUPR_OUT) are included in the CIFAR-10 vs (SVHN, CIFAR-100, LSUN, ImageNet) OOD detection experiments in Table 10 (see Global Response PDF). The result is consistent and we will add the full results in the final manuscript.
> ___
> **[Q6]** Improving the ability of machines to identify novel information could potentially lead to privacy concerns, as more accurate models might be misused for surveillance or unauthorized data collection. In high-stakes applications, an over-reliance on automated novelty detection systems that might still make mistakes could lead to serious consequences.\
> **[A6]** We appreciate the reviewer for elaborating on the limitations of our method applied in real-life scenarios. We will incorporate this discussion into the limitation section in the final draft.
> ___
> **References**\
> [1] Unsupervised out-of-distribution detection with diffusion inpainting, ICML 2023\
> [2] The unreasonable effectiveness of deep features as a perceptual metric, CVPR 2018\
> [3] Benchmarking neural network robustness to common corruptions and perturbations, ICLR 2019

---

> > ### Comment · Reviewer_9KBe · 2023-08-18
> >
> > I acknowledge I have read the rebuttal.

---

> > > ### Author Response · Authors · 2023-08-18
> > >
> > > We thank you for your time to review our paper and read our rebuttal.
> > >
> > > We sincerely appreciate (i) your acknowledgment of our work to **“present significant contribution to the field”** and (ii) your extensive suggestions to **“strengthen our contributions and appeal to the broader audience”**. As we strongly believe that we have successfully addressed all questions and concerns raised by the initial review, we politely ask you to consider feedback on our responses or update the score.
> > >
> > > Since we still have time remaining for the author-reviewer discussion period, please feel free to ask any further questions.
> > >
> > > Thanks.
> > >
> > > Best, \
> > > Authors

---

### Official Review · Reviewer_ZGie · 2023-07-03

**Soundness:** 3 good
**Presentation:** 2 fair
**Contribution:** 3 good
**Rating:** 5
**Confidence:** 4

**Summary:**

This paper presents a novel approach to detecting and handling novelty in data using a generative diffusion model, with a specific focus on addressing biased backgrounds. The proposed method aims to transform noisy samples into perfect ones by leveraging the capabilities of the diffusion model. The central idea is that by reversing the effects of noise on an inlier sample (x+noise), the resulting output will closely resemble the original sample (X). Conversely, if X were to be treated as an out-of-distribution (OOD) sample, the diffusion model trained on similar OOD backgrounds could project the samples onto a background resembling the inlier data.

Unlike traditional methods, such as auto-encoders that rely on reconstruction error, this approach emphasizes the significance of high and meaningful reconstruction errors for anomaly detection. Since the primary focus is on the foreground, which constitutes a small portion of the images, the reconstruction error becomes a more reliable indicator of anomalies.

To validate the effectiveness of the proposed method, the authors conducted evaluations on various datasets, including CIFAR-10, CIFAR-100, SVHN, and ImageNet. Comparative analyses were performed against existing generative models commonly employed for novelty detection.

**Strengths:**


Recently, there has been growing concern that existing methods for image classification, which are generally not specific to one-class classification or novelty detection tasks, face significant challenges. Additionally, studies in the field have highlighted the problem of bias in the background for novelty detection. In response, this paper addresses this ongoing challenge by proposing a solution.

The paper leverages the reverse step of the diffusion model to restore noisy outliers, with the goal of aligning the background of these recovered samples more closely with the inlier samples. This approach is intriguing and tackles a specific aspect of the problem.

The results demonstrate the effectiveness of the proposed idea.

**Weaknesses:**

One weakness of this paper is that the main idea of using projection and reconstruction error for novelty detection is not novel. There have been previous papers, such as "Adversarially One-Class Classifier for Novelty Detection" (CVPR 2018), that have presented similar ideas but with different approaches. Thus, the high-level novelty of the proposed method is questionable. However, the method is novel.

Furthermore, the paper claims the method is less biased towards the background. However, the datasets used for evaluation, such as CIFAR and SVHN, are not known for having extreme biases toward the background. To address the bias problem effectively, evaluating the method on datasets that exhibit strong biases in the background would be more appropriate, such as "Hard ImageNet: Segmentations for Objects with Strong Spurious Cues." Other papers specifically tackle the issue of spurious correlations and out-of-distribution detection, such as "On The Impact Of Spurious Correlation For Out-of-distribution Detection" by Yifei Ming (2021). It would be valuable to compare the results of the proposed method with those papers that specifically address the bias problem rather than just comparing against other generative models.

Moreover, the proposed method may fail in cases where the OOD samples have backgrounds that are very similar to the inliers but differ in the main concept or foreground. In such scenarios, the background would be ideally recovered using the proposed method, leading to false positives where OOD samples are detected as inliers.



At the moment there are a lot of methods that try to handle the same problem to the reconstruction error can be a reliable score for novelty detection for example, see the Mem-autoencoder paper for novelty or anomaly detection.


Addressing these weaknesses and providing a more thorough comparison with relevant papers and datasets would strengthen the overall contribution of the proposed method.




**Questions:**

The idea and problem mentioned in this paper are interesting. However, the authors need to consider the following points: (1) They should review documents in the field of fairness for anomaly detection and those that attempt to mitigate bias. (2) There are benchmarks for evaluating methods biased against the background, such as Hard ImageNet. While these datasets may not be explicitly designed for out-of-distribution (OOD) detection, evaluating the proposed method on such data would be beneficial. Although CIFAR-10 and CIFAR-100 not have meaningful background concepts, additional evaluations on datasets with more pronounced background biases would provide valuable insights. (3) The comparison with existing methods needs improvement."

---

> ### Author Rebuttal · Authors · 2023-08-09
>
> Dear Reviewer ZGie,
>
> We sincerely thank you for your helpful feedback and insightful comments. In what follows, we address your concerns one by one.
> ___
> **[Q1]** One weakness of this paper is that the main idea of using projection and reconstruction error for novelty detection is not novel. There have been previous papers, such as "Adversarially One-Class Classifier for Novelty Detection" [1], that have presented similar ideas but with different approaches. Thus, the high-level novelty of the proposed method is questionable. \
> **[A1]** We ​​politely disagree with the reviewer’s opinion: our main idea is to use the **“recursive projection”** and the **“regret term”**, and it is clearly different from the previous works, e.g., [1]. As we discussed in Section 3.2, a simple reconstruction approach like [1] may fail to capture semantic changes when background information is dominant. Hence, our idea plays a crucial role in novelty detection as we empirically verified in our ablation experiment (Table 3). We strongly believe that our work is a non-trivial extension of the previous reconstruction-based approaches, i.e., our idea is novel and effective compared to them. We respectfully ask you to reconsider your position on our high-level novelty.
> ___
> **[Q2]** Furthermore, the paper claims the method is less biased towards the background. However, the datasets used for evaluation, such as CIFAR and SVHN, are not known for having extreme biases toward the background. To address the bias problem effectively, evaluating the method on datasets that exhibit strong biases in the background would be more appropriate, such as "Hard ImageNet: Segmentations for Objects with Strong Spurious Cues." ...\
> **[A2]** Thank you for the valuable suggestion on experiment setups. As you suggested, we validate the effectiveness of our method on a spurious out-of-distribution (OOD) benchmark, ColorMNIST [2], presented by Ming et al. [3] where digits are given as semantic information. In this benchmark, different digits with the same background color are given as spurious OOD samples. Furthermore, in-distribution digits and background information are highly correlated (r=0.45) in the training setup. As we reported in Table 6 (see Global Response PDF), our method significantly outperforms [3] even without using label information. This is because our method relies on generative modeling which aims to learn every pixel information rather than collapsing into a shortcut solution. It is also worth noting that the methods [2-3] that address the bias problem in classifiers cannot be directly applied to unsupervised OOD detection since the above methods require label information.
>
> We believe this discussion further strengthens our contribution, especially for various practical and challenging scenarios. We will incorporate this experimental result and discussion into the final manuscript.
> ___
> **[Q3]** Moreover, the proposed method may fail in cases where the OOD samples have backgrounds that are very similar to the inliers but differ in the main concept or foreground. In such scenarios, the background would be ideally recovered using the proposed method, leading to false positives where OOD samples are detected as inliers.\
> **[A3]**  We clarify that such a scenario is not our failure case because our detection score is based on the distance between a test image and its projection. If an out-of-distribution (OOD) sample has background statistics that are very similar to the inliers but differ in the main concept, then our projection of the OOD sample becomes similar to an inlier one, which leads to a large distance between the OOD sample and its projection due to their semantic distance. In contrast, an in-distribution sample has a small distance because its projection has similar semantic information.
>
> This claim is also supported by our experimental results. First, although the ColorMNIST dataset [2] presented in [A2] includes such OOD samples (i.e., different digits but the same background colors), our method achieves near-perfect OOD detection performance as reported in Table 6 (see Global Response PDF). In addition, we think that CIFAR-10 vs CIFAR-100 is another representative task of discriminating OOD samples with similar backgrounds. In this task, we outperform the previous best diffusion-model-based method, LMD [4], by a large margin. These experimental results verify that our method can detect OOD samples that have similar backgrounds to the inliers.
> ___
> **[Q4]** Comparison against Mem-autoencoder paper.\
> **[A4]** We appreciate you introducing the related work, Mem-autoencoder (MemAE) [5]. For the comparison against [5], we experiment with Projection Regret and other competitive baselines in the one-class classification task on CIFAR-10 and report their results in Table 7 (see Global Response PDF). Our method greatly outperforms the reconstruction-based baselines, including MemAE, across all classes. For example, Projection Regret outperforms MemAE by 25\%.
> ___
> **[Q5]** Providing a more thorough comparison with relevant papers and datasets would strengthen the overall contribution of the proposed method.\
> **[A5]** We appreciate the reviewer for the introduction of related fields and methods. We will include the related research on spurious correlations [2] and reconstruction-error-based novelty detection methods [1,5] further in the final manuscript.
> ___
> **References**\
> [1] Adversarially learned one-class classifier for novelty detection, CVPR 2018\
> [2] Invariant Risk Minimization, Arxiv 2019\
> [3] On the impact of spurious correlation for out-of-distribution detection, AAAI 2022\
> [4] Unsupervised out-of-distribution detection with diffusion inpainting, ICML 2023\
> [5] Memorizing normality to detect anomaly: memory-augmented deep autoencoder for unsupervised anomaly detection, ICCV 2019

---

> > ### Comment · Reviewer_ZGie · 2023-08-15
> > **upgrade my evaluation.**
> >
> > Thank you for your response. While many of my concerns have been addressed by the authors, there seems to be a lingering issue regarding (1) the novelty of the approach. I concur with the authors' assertion that their method diverges from a simple reconstruction-based model, instead employing a more refined approach with superior techniques. Additionally, (2) the computational costs associated with this method are notably higher than their predecessors. However, I still find the concern regarding novelty valid. Although the method is functional (and the perception of novelty can be pretty subjective), I have decided to revise and upgrade my evaluation.

---

> > > ### Author Response · Authors · 2023-08-16
> > >
> > > Thank you for your additional comments. We further experiment with reducing the computational cost of our algorithm. We add the new results in the official comment section. We will incorporate the discussion and reflect on other reviewers' feedback in the final manuscript.

---

### Official Review · Reviewer_mygo · 2023-07-06

**Soundness:** 3 good
**Presentation:** 3 good
**Contribution:** 2 fair
**Rating:** 5
**Confidence:** 3

**Summary:**

This paper proposes Projection Regret (PR) to mitigate the bias of background information for novelty detection. As an effective perceptual distance, it is able to detect abnormality by reducing the effect of dominant background using recursive projections. Experimental results show the effectiveness of the proposed novelty detection framework.

**Strengths:**

1. Good motivation to explain the background bias problem.
2. well written and easy to follow this paper.
3. The effectiveness of Projection Regret.


**Weaknesses:**

My main concerns lie in the following two aspects: (1) About the detection running time. Diffusion models are used in the proposed method, which leads to the increase of inference time when detecting the abnormality of test images. (2) It lacks experimental comparison on other practical industry product detection tasks, such as MVTecAD and BTAD benchmark, which seem to be more challenging to evaluate the effectiveness of the proposed method.

**Questions:**

Refer to the Weaknesses.

**Limitations:**

Yes.

---

> ### Author Rebuttal · Authors · 2023-08-09
>
> Dear Reviewer mygo,
>
> We sincerely thank you for your helpful feedback and insightful comments. In what follows, we address your concerns one by one.
> ___
> **[Q1]** About the detection running time. Diffusion models are used in the proposed method, which leads to an increase of inference time when detecting the abnormality of test images.\
> **[A1]** During inference, our method (Projection Regret, PR) is very efficient because PR only requires three projections and each projection can be computed by one forward pass using a consistency model [1] that enables one-shot generation, as described in Algorithm 1. The second-best baseline, LMD [2], on the other hand, requires multiple (sequential) sampling steps for inpainting. As a result, PR is 1.8x faster than LMD.
>
> ---
> **[Q2]** It lacks experimental comparison on other practical industry product detection tasks, such as MVTecAD and BTAD benchmark, which seem to be more challenging to evaluate the effectiveness of the proposed method. \
> **[A2]** We first note that industrial anomaly detection (IAD) like MvTecAD and BTAD benchmarks are not within our scope since they differ from our target task: unsupervised novelty detection (UND), also known as out-of-distribution (OOD) detection. To be specific, UND aims to detect OOD (i.e., novel or unseen) semantic categories/shapes, while the IAD task aims to detect a localized defect in a pixel-wise manner. Therefore, their sample distributions are also significantly different. For example, OOD samples of the IAD task only differ in the small localized region against the ID samples, but OOD samples of the UND task may differ in shapes, background, and so on. Due to the aforementioned reasons, the UND and IAD tasks have been investigated in different research directions, e.g., formulating why the novelty detection model overfits to non-semantic information in UND [3] and formulating abnormality score in local pixels in IAD [4,5].  We think that IAD is an interesting extension of our work and leave the extension for future work.
> ___
> **References** \
> [1] Consistency models, ICML 2023 \
> [2] Unsupervised out-of-distribution detection with diffusion inpainting, ICML 2023 \
> [3] Novelty detection via blurring, ICLR 2020 \
> [4] Towards total recall in industrial anomaly detection, CVPR 2022 \
> [5] Pushing the limits of few-shot anomaly detection in industry vision: Graphcore, ICLR 2023

---

> > ### Comment · Reviewer_mygo · 2023-08-17
> > **After the rebuttal**
> >
> > Thanks for the authors' response. In the rebuttal. the authors cannot well address my concerns, w.r.t. running time and comparison problems. Although the proposed method seems to be novel, addressing my concerns is important to further evaluate the effectiveness of the proposed method. In the evaluated datasets, I think they are less biased toward the background. I will decrease my initial score to borderline accept.

---

> > > ### Author Response · Authors · 2023-08-17
> > >
> > > Dear Reviewer mygo,
> > >
> > > We appreciate your additional feedback on the rebuttal. We would like to further discuss with you about your concerns, especially on (1) the running time and (2) the evaluated datasets.
> > >
> > > ___
> > >
> > > **[Q1]** About the running time \
> > > **[A1]** In the following official comment (https://openreview.net/forum?id=3qHlPqzjM1&noteId=lKGqSQ3GLx), we summarize that our method can show significant acceleration (8.7x) compared to the second-best baseline, LMD [1]. Hence, we want to emphasize that our method is efficient compared to the existing out-of-distribution (OOD) detection methods that use diffusion models.
> > > ___
> > >
> > > **[Q2]** Although the proposed method seems to be novel, addressing my concerns is important to further evaluate the effectiveness of the proposed method. In the evaluated datasets, I think they are less biased toward the background. \
> > > **[A2]** We note that Reviewer ZGie and Reviewer SZQH questioned similar concerns on the evaluation. As such, in the rebuttal (https://openreview.net/forum?id=3qHlPqzjM1&noteId=gDN0zwDfDL), we performed the new experiment where the OOD dataset has the same background information as the in-distribution dataset. In such a dataset, our method shows significant performance gain (see Table 6 on Global Response PDF). Furthermore, the OOD detection methods trained in the CIFAR-10 dataset show underwhelming detection performance in detecting the CIFAR-100 OOD dataset where both methods share similar background information. Our method greatly increases the performance in such a dataset (see Table 1). It is also worth noting that reviewer ZGie acknowledged that our rebuttal addressed the concerns (update: both reviewers acknowledge our rebuttal).
> > >
> > > Furthermore, OOD detection and industrial anomaly detection (IAD) have been researched in separate directions. We strongly note that the application of OOD detection methods in the IAD benchmark is not a standard practice. For example, every baseline paper [1-6] that we compared does not experiment on such datasets. Nevertheless, in the rebuttal, we scaled our methods to the large-scale dataset (e.g. LSUN) and observed significant performance gain (see Table 8 in Global Response PDF). We ask the reviewer to keep this in mind and reconsider your concerns about the evaluation of such datasets.
> > >
> > > ___
> > >
> > > In summary, we politely ask the reviewer to discuss 1) why our evaluated datasets are less biased toward the background even though we experimented on the dataset with the same background information, 2) why comparison against the IAD dataset is the issue where we also experimented on the large-scale dataset and none of the baseline methods did, and 3) why our computation time is the issue even though we observed significant gain on the computational cost compared to the second-best diffusion-based OOD detection method.
> > >
> > > We hope this discussion would clarify your concerns and please feel free to ask any further questions.
> > >
> > >
> > > ___
> > >
> > > **References**\
> > > [1] Unsupervised out-of-distribution detection with diffusion inpainting, ICML 2023\
> > > [2] Input complexity and out-of-distribution detection with likelihood-based generative models, ICLR 2020\
> > > [3] Likelihood regret: an out-of-distribution detection score for variational autoencoder, NeurIPS 2020\
> > > [4] Multiscale score matching for out-of-distribution detection, ICLR 2021\
> > > [5] VAEBM: a symbiosis between variational autoencoders and energy-based models, ICLR 2021\
> > > [6] Guiding energy-based models via contrastive latent variables, ICLR 2023

---

### Author Rebuttal · Authors · 2023-08-09

Dear reviewers and ACs,

We sincerely appreciate your valuable time and effort spent reviewing our manuscript.

As reviewers highlighted, our work aims at an important problem **(Reviewer ZGie, 9KBe, SZQH)** with an interesting/novel method **(Reviewer ZGie,9KBe)**, strong empirical results **(Reviewer mygo, ZGie, 9KBe, SZQH)**, and a well-written/easy-to-follow writeup **(Reviewer mygo, 9KBe, SZQH)**.

We appreciate your constructive comments on our manuscript. We have carefully addressed the comments with the following additional discussions and experiments:
- **[Reviewer ZGie, SZQH]** Performance of Projection Regret on spurious OOD datasets with the same background information (Table 6).
- **[Reviewer ZGie]** Performance of Projection Regret in CIFAR-10 one-class classification benchmark (Table 7).
- **[Reviewer 9KBe, SZQH]** Performance of Projection Regret applied to larger diffusion models with larger dataset size (Table 8).
- **[Reviewer 9KBe]** Robustness analysis on Projection Regret against common corruptions (Table 9, Figure 5).
- **[Reviewer 9KBe]** Performance of Projection Regret measured in alternative metrics (Table 10).
- **[Reviewer SZQH]** Sensitivity analysis of Projection Regret and the motivation experiment extended to multiple OOD datasets (Figure 6).

We hope our response sincerely addresses all the reviewers’ concerns.

Thank you very much.

Best regards, \
Authors.

---

### Author Response · Authors · 2023-08-16
**A gentle reminder**

Dear Reviewers and ACs,

We again appreciate your valuable feedback on our submission. We have addressed your concerns and questions (e.g., scalability to the large-scale dataset, robustness, dataset with spurious correlation, and alternative metrics) throughout various experiments. To further discuss remaining concerns during this limited rebuttal period, we first (1) assert the motivation and virtue of our method and also (2) elaborate on the **computational cost** of our method through **new experimental results**.

First, we propose an out-of-distribution detection (OOD) method via diffusion models (or consistency models) since they are widely used, and able to generate photorealistic images, hence may show potential in various downstream tasks. However, existing methods either show underwhelming OOD performance or require a high computational cost due to their requirement of multiple sampling. We show improvement on both sides: competitive performance in various domains (see Table 1, Table 2, Table 7 in Global Response PDF) and gain in computational cost.

We further show that our method is robust while reducing computational costs for practical applications. For example, in Figure 4 (right), we show that even with 10x fewer computations, the AUROC of CIFAR-10 vs CIFAR-100 OOD detection performance decreases by less than 0.01. In such a hyperparameter, we take 0.081s/sample in wall-clock time and hence 8.7x faster than the computational cost of LMD. We report the OOD detection performance and the computational cost of the original version and the fast version compared to the LMD in the table below.

___
|Method  |SVHN |CIFAR-100|LSUN  |ImageNet|computational cost(sec/sample)|
|--------|-----|---------|------|--------|------------------|
|LMD [1] |0.979|0.620    |0.734 |0.686   |0.705             |
|PR (ours, fast version)|0.992|0.767    |0.828 |0.806   |0.081             |
|PR (ours)      |0.993|0.775    |0.837 |0.814   |0.395             |

Again, we truly appreciate your effort and time to review this work!

Best, \
Authors


**References**\
[1]  Unsupervised out-of-distribution detection with diffusion inpainting, ICML 2023

---

### Author Response · Authors · 2023-08-21

Dear AC and reviewers,

We appreciate the thoughtful feedback to improve the paper. While all the reviewers are on the positive side of the work, there seems to be an unresolved concern about the **computational cost** of our algorithm.

While our first rebuttal addressed such concerns using a small-sized dataset (i.e., 32x32), we here further verify that our method is still efficient even in a large-scale dataset, e.g., LSUN (256x256), which is a more challenging and practical scenario.

Since the main hyperparameter for our computation cost is the ensemble size (i.e., $n_{\alpha}$ and $n_{\beta}$), we experiment with the performance of Projection Regret in smaller ensemble sizes. Specifically, we set $n_{\beta}=1$ and further test the performance under the various choices of the ensemble size $n_{\alpha} \in  ${5, 10, 20, 30, 40}$ $.

As shown in the table below, our method can be scaled down to $n_{\alpha}=5, n_{\beta}=1$ which takes 29x faster (i.e., 0.53s/sample vs 15.4s/sample) than the second best baseline (LMD [1]). We strongly believe this shows the practical availability of our method and hope to clarify the reviewer’s concern about the computational cost.

|$n_{\alpha}$ |bridge |church|classroom|
|--------|-----|---------|------|
|LMD [1] |0.386|0.379|0.336|
|PR (original) |0.830|0.772|0.696|
|40|0.833|0.773|0.699|
|30|0.827|0.764|0.691|
|20|0.830|0.766|0.688|
|10|0.821|0.770|0.691|
|5 |0.824|0.740|0.692|

Again, we truly appreciate your effort and time to review this work!

Best, \
Authors


**References**

[1] “Unsupervised out-of-distribution detection via diffusion inpainting”, ICML 2023

---

### Decision · Program_Chairs · 2023-09-21

**Decision:**

Accept (poster)

**Comment:**

The proposed work on novelty detection by reducing background bias has received no strong support for acceptance, but at the same time none of the reviewers is against acceptance. AC has read the paper as well and believes the authors have adequately addressed the reviewer concerns on background bias, concerning computational efficiency of the diffusion-based approach, and a lacking experimental justification on real-world datasets. While the work may leave something to be desired, for example a more thorough qualitative analysis of background bias, e.g. using the datasets suggested by Reviewer mygo, the current paper is of sufficient interest. Discussion with the SAC further strengthened AC in recommending acceptance, given that all promised discussions and experiments will be added to the camera-ready version.